# Does problem-based learning improve patient empowerment and cardiac risk factors in patients with coronary heart disease in a Swedish primary care setting? A long-term prospective, randomised, parallel single randomised trial (COR-PRIM)

Christina Andreae [1,2] Pia Tingström,[1] Staffan Nilsson,[1,3] Tiny Jaarsma,[1,4] Nadine Karlsson,[1] Anita Kärner Köhler[1]

For numbered affiliations see end of article.

**Correspondence to**
Dr Christina Andreae;
christina.andreae@liu.se

## ABSTRACT

**Objectives** To investigate long-term effects of a 1-year problem-based learning (PBL) on self-management and cardiac risk factors in patients with coronary heart disease (CHD).

**Design** A prospective, randomised, parallel single centre trial.

**Settings** Primary care settings in Sweden.

**Participants** 157 patients with stable CHD completed the study. Subjects with reading and writing impairments, mental illness or expected survival less than 1 year were excluded.

**Intervention** Participants were randomised and assigned to receive either PBL (intervention) or home-sent patient information (control group). In this study, participants were followed up at baseline, 1, 3 and 5 years.

**Primary and secondary outcomes** Primary outcome was patient empowerment (Swedish Coronary Empowerment Scale, SWE-CES) and secondary outcomes General Self-Efficacy Scale (GSES), self-rated health status (EQ-VAS), high-density lipoprotein cholesterol (HDL-C), body mass index (BMI), weight and smoking. Outcomes were adjusted for sociodemographic factors.

**Results** The PBL intervention group resulted in a significant improved change in SWE-CES over the 5-year period (mean (M), 39.39; 95% CI 37.88 to 40.89) compared with the baseline (M 36.54; 95% CI 35.40 to 37.66). PBL intervention group increased HDL-C level (M 1.39; 95% CI 1.28 to 1.50) compared with baseline (M 1.24; 95% CI 1.15 to 1.33) and for EQ-VAS (M 77.33; 95% CI 73.21 to 81.45) compared with baseline (M 68.13; 95% CI 63.66 to 72.59) while these outcomes remained unchanged in the control group. There were no significant differences in BMI, weight or scores on GSES, neither between nor within groups over time. The overall proportion of smokers was significantly higher in the control group than in the experimental group.

**Conclusion** One-year PBL intervention had positive effect on patient empowerment, health status and HDL-C at a

## STRENGTHS AND LIMITATIONS OF THIS STUDY

⇒ The majority of participants remained in the study over the 5 years follow-up.
⇒ Baseline measurement was well balanced for both groups regarding sociodemographic and study variables.
⇒ The problem-based learning (PBL) intervention was held by a small group of specialist nurses who were advanced trained in PBL.
⇒ Participants were most retired, were men and lived in suburban area, which need to be considered in the implementation of PBL in cardiac primary care settings.

5-year follow-up compared with the control group. PBL education aiming to improve patient empowerment in cardiac rehabilitation should account for sociodemographic factors.

**Trial registration number** NCT01462799.

## INTRODUCTION

Nearly 126 million people globally suffer from coronary heart disease (CHD), the leading cause of death worldwide.[1 2] Important treatment goals include slowing down the underlying atherosclerosis process by targeting hypertension, high cholesterol, diabetes, overweight, tobacco use, alcohol consumption, physical inactivity and poor diet.[3] Cardiac rehabilitation in postmyocardial infarction improve lifestyle habits such as physical activity and dietary intake, and has also shown positive effects on blood lipids, blood pressure and smoking.[4–6] In a small randomised trial, cardiac rehabilitation

programme for patients with myocardial infarction or who had undergone coronary artery by-pass graft surgery (CABG), focusing on stress management, physical exercise and dietary intake, reduced morbidity and hospitalisation for CHD.[7] Attendance in cardiac rehabilitation programme has long-term effects on survival among patients who have undergone CABG with reduced 10-year all-cause mortality.[8] Cardiac rehabilitation enables patients to make lifestyle changes that are important for maintaining health. Cardiac rehabilitation consists of multidisciplinary interventions where patients are offered tailored education and counselling with the goals of improving health behaviour for sustainable secondary prevention.[3] Several studies have investigated the effect of interventions such as nurse telephone follow-up or group education, and described the effects on illness perception, self-efficacy, behaviour change and cardiac risk factors.[9–11] The effects of several programme on cardiac risk factors or behaviour were found from 3 months up to 2 years. Patients' attendance of cardiac rehabilitation appears to be low, and risk factors remain or even deteriorate from the first to second cardiac event.[12–14] For example, persistent smoking is common after discharge from hospital, and intentions to quit smoking in the near future remain low. This regards also overweight and physical activity after a cardiac event, many patients do not lose weight and nearly half of patients perform no physical activity and do not make changes to their physical status.[13] Thus, primary care has great challenges motivating patients to achieve healthy lifestyles.

Empowering patients to take control over the disease, and making them aware of factors that affect illness, have positively impacted on health outcomes in chronic diseases.[15] Problem-based learning (PBL) is a method that empowers patients to become aware of how to reduce risk factors in chronic diseases.[16] PBL is a cognitive educational model that promotes self-learning where critical and reflective thinking are important components of learning outcomes and enhanced self-management. PBL is process-oriented, meaning that learning skills are developed through active, creative and cognitive processes. Previous knowledge is placed in the light of real situations where knowledge is developed in a context together with others, all led by a trained PBL tutor.[17] A focus group intervention with problem solving intervention in a cardiac population showed that an 8-week focus group sessions resulted in significant improvements of stress management, dietary intake and physical activity.[18] The first results from a 1-year follow-up in a Coronary Heart Disease in Primary Care Study (COR-PRIM) showed no differences in patient empowerment or self-efficacy between PBL intervention and patients who received home-sent patient information (controls). However, significant differences in secondary outcomes—that is, body mass index (BMI), body weight and high-density lipoprotein cholesterol (HDL-C)—were found between the groups, with the PBL group performing favourably.[19]

Although cardiac rehabilitation in Sweden has recently been assessed to be high-quality[20] participation remains low. Only a few interventions incorporating a holistic view of health compromising psychological and social aspects of health behaviours have been performed on the group level. PBL may empower self-management. This study is elaborated in accordance with the COR-PRIM study basic aim, which was to discover whether PBL provided in primary healthcare for 1 year has long-term effects on patient empowerment and self-care, assessed at baseline and, at 1, 3 and 5 years after randomisation. In this 5-year and final assessment, we wanted to identify if the findings in the 1-year follow-up remained or changed.[19] By this performance, this article is examining the sustainability of the effects by PBL, which to our knowledge has not been performed before.

Thus, the purpose of this study was to investigate the long-term effects on patient empowerment, self-efficacy, health status and cardiac risk factors in patients with CHD of a 1-year PBL intervention in primary care, compared with home-sent patient information.

## METHODS
### Design, sample and procedure
The COR-PRIM study was a prospective, randomised, parallel and single-centre trial designed to investigate whether 1 year of a PBL programme had long-term effects on patient empowerment, self-efficacy and cardiac risk factors.[17] The study protocol is registered at ClinicalTrial. gov, with the registration number NCT01462799. Patients diagnosed with CHD verified by percutaneous coronary intervention (PCI) or coronary artery by-pass surgery (CABG) and CABG+PCI or myocardial infarction within 6–12 months prior to the intervention and who had previously completed cardiac rehabilitation were eligible for the study. Additional criteria were that patients were stable in their heart disease, pharmacologically optimised in the last month before inclusion and, if applicable had completed cardiac school in the clinic. Patients with impaired ability to communicate and read in Swedish, verified psychiatric illness or short expected survival (of less than 1 year) were excluded.[17] The recruitment started in November 2011 and assessments of primary and secondary outcomes were collected until November 2019. Nurses at outpatient cardiac clinics identified eligible patients through medical records. These patients were then invited to participate in the study at a regular nurse-led cardiac follow-up visit. Patients received oral and written study information from a research assistant, and written consent was obtained from those interested in participating (online supplemental material 1).

### The study randomisation and intervention
In COR-PRIM, a total of 157 patients were randomised and assigned to either PBL or home-sent patient information (control group). The randomisation procedure was based on block randomisation with 18 study numbers

based on a minimum of 12–18 study participants. Block randomisation was performed using sealed and unmarked envelopes. An assistant, blinded to the randomisation and located outside the research setting, randomly allocated patients to either the PBL group (experimental) or home-sent patient information (control group) at a 1:1 ratio.[17] Nurses who performed the PBL intervention were not blinded, as the intervention was based on PBL, which is different from regular care. The PBL intervention consisted of group learning sessions in primary care led by nurses who had advanced training in leading PBL. Tutoring groups consisted of 6–9 participants and participated in 13 sessions (2 hours per meeting) over 1 year. This was based on the multiple of areas to discuss for lifestyles changes in CHD. For further information about the PBL intervention, see online supplemental material 2. Lifestyles changes require work over time and to be feasible in the patient's life, a long training is required. The PBL model has been validated as feasible in clinical settings.[21] Patients randomised to the control group received home-sent patient information reflecting a cognitive intervention, but did not participate in critical or reflective learning. Group effects were assessed after 11 sessions over 1 year.[17]

### Patient and public involvement
There were no patient or public involvement in the study.

### Sample size
Sample size was calculated based on the primary outcome variable, that is, patient empowerment. The expected means for the Swedish Coronary Empowerment Scale (SWE-CES) were 30 and 36 for the control group and the PBL group, respectively. With an estimated significance level of $\alpha=5\%$ and a power of $1-\beta=80\%$, the sample size resulted in a minimum of 63 participants in each group.[17] Analyses conducted for the 5-year follow-up period was n=72 (PBL) and n=71 (control group) (figure 1).

### Primary and secondary outcome measures
Patient empowerment was the study's primary outcome and was assessed at baseline, 1, 3 and 5 years (ie, baseline, time 1, time 3 and time 5, respectively). Secondary outcomes included general self-efficacy while five secondary outcomes were added post hoc and these were health status, HDL-C, BMI, weight and smoking and were assessed at baseline, 1, 3 and 5 years.

### Measurement
#### Sociodemographic variables and covariates
Sociodemographic variables consisting of age, sex, education level, marital status, place of residence, employment and smoking were determined by self-reported questionnaires. The covariates consisted of age, sex, education level and marital status.

#### Self-reported measures
The SWE-CES measures how patients achieve goals, overcome barriers for goals achievement and use the

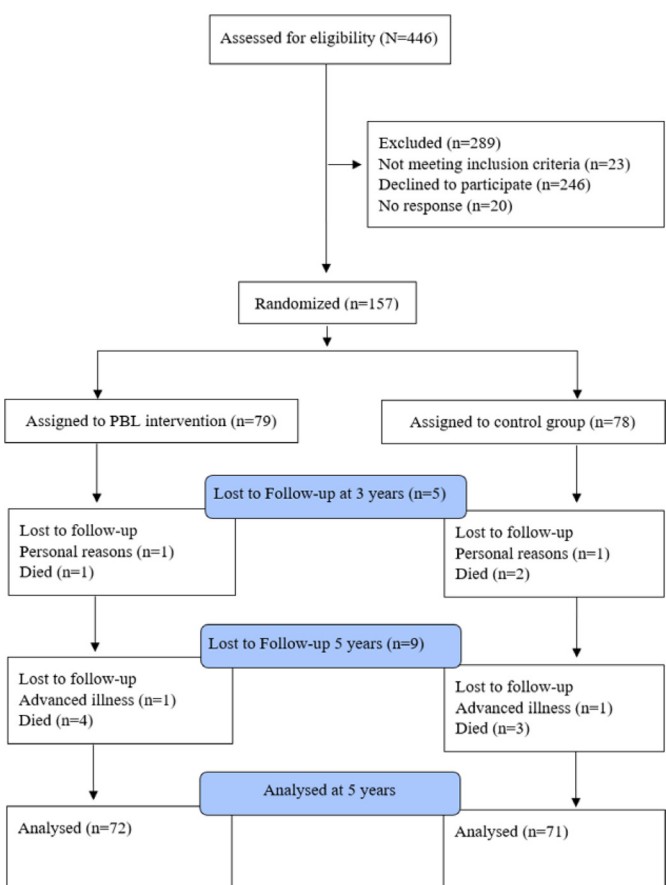

**Figure 1** Flow chart on study participation.

strategies necessary to make self-care choices. The strategies include dimensions of coping in managing disease, stress and dissatisfaction and also readiness to make health changes. The instrument contains 10 items that have 5 response alternatives ranged by Likert type options from 1 to 5. The total score ranges between 10 and 50, where high scores imply high level of patient empowerment. Four subscales measure different aspects of patient empowerment.[22] In this study, the total scores of the SWE-CES scale were used. Internal consistency reliability indicated acceptable values with Cronbach's alpha 0.751.[23]

Self-efficacy was measured by using the General Self-Efficacy Scale (GSES). The instrument consists of 10 items that measure a person's belief in their own ability to implement behavioural changes in order to reduce risk factors for unhealthy lifestyles. Items are rated on a four-point Likert type scale ranging from 1 (not at all true) to 4 (exactly true). The total score ranges from 10 to 40, where a high total score indicates higher general self-efficacy.[24] Self-efficacy has been used to evaluate interventions to strengthen the self-care capacity of patients with various chronic diseases such as diabetes[25 26] and patients with CHD.[27–29] It has been translated and psychometrically evaluated among a general Swedish population.[30] In this study, internal consistency reliability indicated acceptable values with Cronbach's alpha 0.914.

## Health status

Self-rated health was assessed using EuroQol Visual Analogue Scale (EQ-VAS). The scale represents patients' overall health status and consists of a score between 0 and 100, where 0 indicates worst imaginable state of health and 100 the best imaginable state of health.[31] EuroQol Visul Analogue Scale (EQ-VAS) has shown acceptable construct validity across populations.[32]

## Cardiac risk factors

HDL-C was assessed using blood samples collected and analysed according to normal clinical routines. Anthropometric measurements of body size included length, weight and BMI. Length and weight were measured in light cloths and with shoes removed. BMI is a widely used clinical measure and is recommended as indicator for defining obesity in adults. It is, furthermore, a reliable anthropometric measure in predicting metabolic syndromes.[1 33] BMI was calculated by dividing weight by the square of height in metres using the formula (weight $(kg)/height$ $(m^2)$).[34] Tobacco use was self-reported at each measure point.

## Statistical analysis

The descriptive statistics included mean SD or frequencies percent (%). Independent sample t-tests or $\chi^2$ tests were used to analyses differences in sociodemographic and data characteristics between groups. Because the occasions (baseline to 5 years) are nested within individuals, we employed a two-level mixed linear model with occasions ('time' at level 1) that are nested within individuals (level 2) to analyse continuous outcomes. The random part of the variance components model is individuals. The normality of continuous study outcomes was assessed using Shapiro-Wilk test. If the outcomes were non-normally distributed, they were log transformed prior to the analysis. Smoking is a binary outcomes and was analysed with a logistic mixed model. The statistical analysis of each study outcome was performed using a mixed linear model with treatment group, age and sex as fixed variables, with occasions nested within individuals, and an interaction of time by treatment group. The interaction (combined effect) between treatment group and time was tested using the likelihood ratio test. In the presence of an interaction, the analysis was stratified by treatment group, and a mixed model with occasions nested within individuals was performed separately for each group. These stratified analyses were adjusted in two models. Model 1 was adjusted for age and sex, and model 2 was, furthermore, adjusted for educational level and marital status. The level of significance used was p<0.05. When the interaction term was not significant at 5% but was significant at 10%, a sensitivity analysis was performed by exploring the results of the analysis stratified by intervention group (PBL and control). The statistical analysis was performed in SPSS V.27 and Stata V.16.0.

## RESULTS

### Sample

Of the initial 446 invited participants, 289 were excluded. Of the latter, 23 did not fulfil the inclusion criteria, 246 declined to participate and 20 did not respond to the invitation. The final study group consisted of 157 participants, of whom 79 were assigned to the PBL group and 78 assigned to the control group. At the 3-year follow-up, there were five drop-outs, two participants (one from the control group and one from the PBL group) had not completed the study for personal reasons, and three participants had died (two from the control group and one from the PBL group), leaving n=77 participants (PBL) and n=75 (control group) at the 3-year follow-up. At the 5-year follow-up, there were nine drop-outs, two participants (one from the control group and one from the PBL group) had advanced in their illness, and seven participants had died (three from the control group and four from the PBL group). There was no statistically significant difference in the drop-out rate between treatment groups as tested by $\chi^2$ test (p=0.980).

Of the total sample of 157 participants, the mean age was 68.7 years±8.5 and the majority were men n=122, 77.7% (table 1). Almost half of the participants lived in suburban areas, and the remaining lived in small towns or the countryside. In total, 113 participants were retired and the remaining were in employment. Almost half of the participants had suffered from myocardial infarction, which occurred at a mean of 284 days before the start of the study.

At baseline, the majority had no symptoms of angina pectoris. Less than half of the participants had hyperlipidaemia. Additional characteristics are available from a previous study.[19] The distribution for continuous outcomes for the PBL and control groups at the 1-year, 3-year and 5-year follow-ups is illustrated with a summary of data including minimum, first quartile, median, third quartile and maximum in boxplots (figure 2). The results of the statistical analysis of study outcomes over time and by intervention are summarised below (table 2). Results of the stratified analysis adjusted for age and sex only gave results similar to the model also adjusted for educational level and marital status. Therefore, table 2 presents only results adjusted for age, sex, educational level and marital status.

### Findings of the intervention of PBL on primary and secondary outcomes

#### Patient empowerment

There was a statistically significant change over time of patient SWE-CES (patient empowerment) from baseline to time 5 (p=0.025) in the total group of participants (table 2).

Additional findings of post hoc sensitivity analysis with interaction test significant at 10% was found for SWE-CES. The interaction between time and group was (p=0.086). This implies that the analysis of SWE-CES stratified by group showed a significant increase in the PBL group as a part of the sensitivity analysis. SWE-CES

**Table 1** Baseline characteristics of the participants in the COR-PRIM study randomised to problem-based learning intervention (PBL) or home-sent patient information (control group)

| | Total sample (N=157) | PBL (n=79) | Control (n=78) | P value |
|---|---|---|---|---|
| Age year, mean (SD) | 68.7 (8.5) | 68.5 (9.2) | 68.9 (7.7) | 0.78* |
| Sex, n (%) | | | | |
| Male | 122 (77.7) | 60 (75.9) | 62 (79.5) | |
| Female | 35 (22.3) | 19 (24.1) | 16 (20.5) | 0.70† |
| Marital status, n (%) | | | | |
| Cohabiting | 115 (74.2) | 60 (75.9) | 55 (72.4) | |
| Living alone | 40 (25.8) | 19 (24.1) | 21 (27.6) | 0.71† |
| Education level, n (%) | | | | |
| Compulsory | 84 (54.2) | 46 (58.2) | 38 (50.0) | |
| Upper secondary | 31 (20.0) | 16 (20.3) | 25 (19.7) | |
| University | 38 (24.5) | 17 (21.5) | 21 (27.6) | 0.43† |
| SWE-CES total score, mean (SD) | 36.84 (5.37) | 36.54 (4.94) | 37.15 (5.81) | 0.49* |
| GSES total score, mean (SD) | 31.30 (5.13) | 31.20 (5.44) | 31.41 (4.82) | 0.80* |
| EQ-VAS, mean (SD) | 70.21 (17.9) | 68.13 (19.5) | 72.44 (15.8) | 0.05* |
| HDL-C, mmol/L, mean (SD) | 1.27 (0.44) | 1.24 (0.39) | 1.30 (0.48) | 0.42* |
| BMI, kg/m2, mean (SD) | 27.10 (4.39) | 27.12 (3.91) | 27.07 (4.85) | 0.95* |
| Smoking, n (%) | 19 (12.1) | 9 (11.4) | 10 (12.8) | 0.81† |

*Independent sample t-test.
† χ2 test.
BMI, body mass index; COR-PRIM, Coronary Heart Disease in Primary Care Study; EQ-VAS, EuroQol Visul Analogue Scale ; GSES, General Self-Efficacy, high score implies high general self-efficacy; HDL-C, high-density lipoprotein cholesterol; SWE-CES, Swedish Coronary Empowerment Scale .

increased significantly at time 3 (M=38.34±5.76, p=0.023) and time 5 (M=39.39±5.23, p<0.001) (table 2) compared with the baseline, independently of positive effects of covariates low education (p=0.041); it did not, however, change significantly over time in the control group. Being a woman did, however, negatively effect SWE-CES (p=0.010).

### Self-rated health
We observed a statistically significant change of EQ-VAS (self-rated health status) from baseline to time 1 (p=0.038). The interaction between time and group was significant at the level of 5% (p=0.022). Therefore, the analysis was stratified by group. EQ-VAS increased significantly at time 1 (M=74.64±18.05, p=0.026), time 3 (M=78.27±14.84, p=0.007) and time 5 (M=77.33±15.52, p=0.031) in the PBL group compared with baseline, independently of negative effects of covariates age (p=0.018), while positive effects of low education increased EQ-VAS by 10 units compared with high education (p=0.011). No significant changes were observed in the control group at time 1 (M=75.30±16.36, p=0.501), time 3 (M=71.89±18.33, p=0.195) and time 5 (M=72.06±17.19, p=0.137) compared with baseline (table 2). The covariate living alone showed a negative effect on EQ-VAS by −7.5 units compared with living together (p=0.039).

### High density lipoprotein cholesterol
There was a statistically significant change of HDL-C from baseline to time 5 (p=0.036) (table 2). The interaction between time and group was significant at alpha level 5% (p=0.016). Therefore, the analysis was stratified by group. HDL-C increased significantly at time 1 (M=1.37 mmol/L±0.43, p=0.003), time 3 (M=1.42 mmol/L±0.55, p<0.001) and time 5 (M=1.39 mmol/L±0.41, p<0.001) in the PBL group compared with baseline, independently of the positive effects of covariates sex (woman) (p=0.002) and negative effects of low education compared high education (p=0.035). There was no significant change in the control group at time 1 (M=1.23±0.36, p=0.781), time 3 (M=1.27±0.37, p=0.655) or time 5 (M=1.27±0.42, p=0.459) compared with baseline (table 2). However, being a woman was positively associated with HDL-C (p=0.026).

### BMI and weight
There was no statistically significant main effect (time or group) for BMI (table 2). There was a significant interaction between time and group at the significance level 10% (p=0.061). Therefore, the analysis was stratified by group. The analysis stratified by group showed a slight decrease from baseline (M=27.12±3.91) after 1 year (M=26.83±3.93, p=0.076), but that did not remain over time. No significant changes over time were observed for

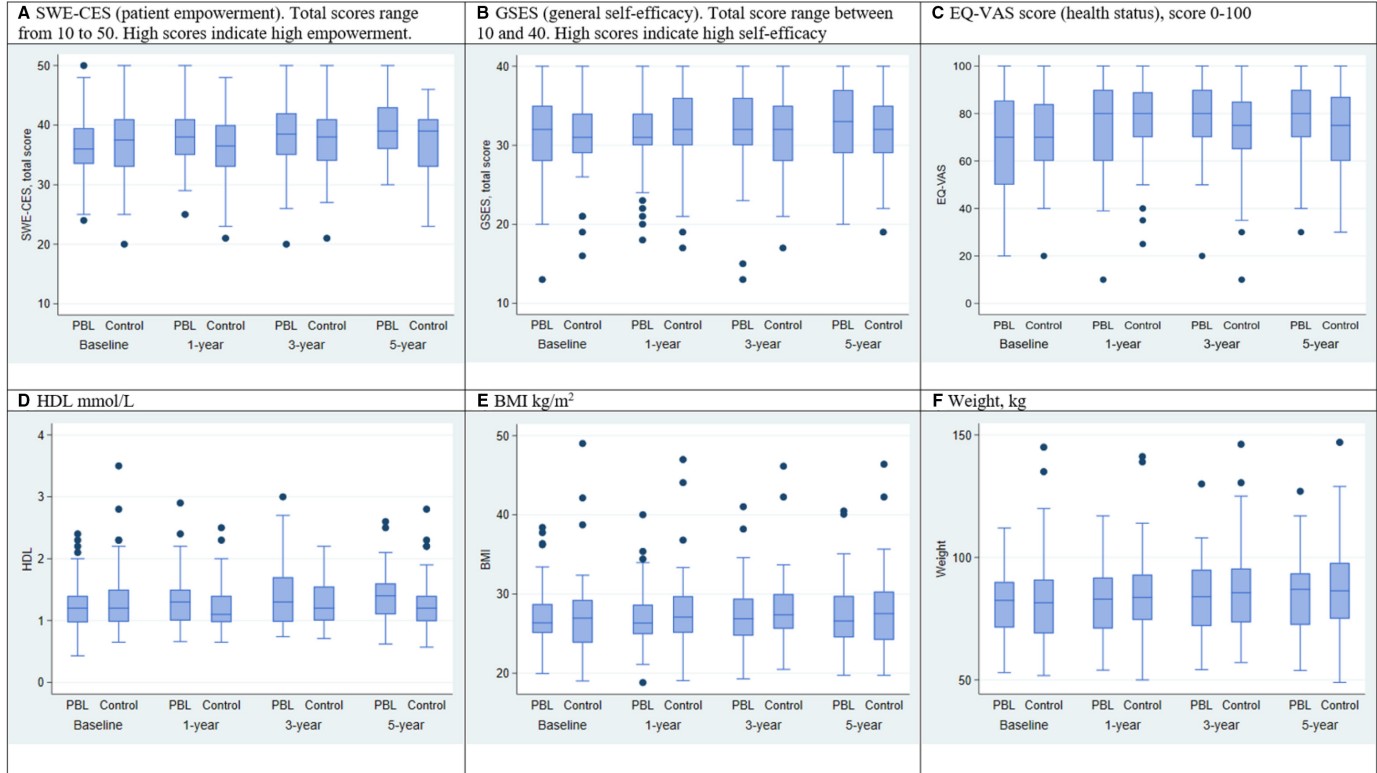

**Figure 2** Distribution of outcome variables in PBL and control groups at baseline, 1-year, 3-year and 5-year follow-up. PBL, problem-based learning.

the control group (table 2). For weight, there was a significant interaction between time and group at alpha level of 5% (p=0.047). Therefore, the analysis was stratified by group. No significant differences were observed between time 1 and 5 in the PBL group compared with baseline. However, the mean weight of patients in the control group at time one increased significantly compared with baseline (p=0.045) (table 2).

### Self-efficacy

For GSES (self-efficacy) there was no significant group effect or change over time (table 2).

### Smoking

There was no significant interaction between time and intervention (p=0.787), but there was a significant group effect (p=0.029) for smoking, showing that the overall proportion of smokers in the control group was statistically significantly higher than the proportion of smokers in the PBL group (table 2).

### DISCUSSION

We investigated the long-term effects of a 1-year PBL intervention in primary care on patient empowerment, self-efficacy, health status and cardiac risk factors in patients with CHD. PBL improved patient empowerment, health status and HDL-C over the 5 years follow-up compared with control group. However, PBL did not result in changes in BMI, weight, smoking or self-efficacy.

We observed significant changes from PBL in patient empowerment and health status over the 5-year follow-up compared with the 1-year follow-up study.[19] It seems possible that it might take a long time to adapt to longstanding behavioural changes. As many studies in this area often end within a year[9 10 35] our study indicates that long-term follow-up is needed to understand the effects on patient behavioural outcomes of interventions using social cognitive theories. Other research shows the need for long-term data on effectiveness of patient education about how to lessen risk factors after CHD.[36]

Cardiac rehabilitation reduces risk factors but may also improve health over time.[37]

A national prospective cohort study with nearly 4500 participants showed that cardiac rehabilitation had positive effects on quality of life up to 1 year after hospital discharge compared with non-participants.[38] In our study, health status improved up to 1 year, and also after 3 and 5 years compared with the control group. PBL allows patients to actively decide what is important to discuss in cardiac rehabilitation, suggesting that PBL might promote control over the disease, leading to better health.

We found that HDL-C improved over the 5 years follow-up. Similar results have been reported in short term follow-up studies on cardiac risk factors using combined education and written information intensive health education sessions and individual support.[39 40] These studies share a focus on education, but we have used PBL in group sessions in a period of 1 year. PBL activates the

**Table 2** Differences in self-management and cardiac risk factors between problem-based learning intervention (PBL) versus home-sent information (control group)

| | Outcome | P value | N | PBL group Mean (SD) or n (%) | P value | N | Control group Mean (SD) or n (%) | P value |
|---|---|---|---|---|---|---|---|---|
| Overall effect of group | SWE-CES | 0.558 | | | | | | |
| Overall effect of time | | 0.113 | | | | | | |
| Time × group | | 0.086* | | | | | | |
| Baseline | | Ref | 76 | 36.54 (4.94) | Ref | 72 | 37.15 (5.81) | Ref |
| Time 1 | | 0.386 | 57 | 37.79 (5.22) | 0.119 | 54 | 36.57 (5.53) | 0.720 |
| Time 3 | | 0.060 | 56 | 38.34 (5.76) | 0.023 | 55 | 37.80 (5.76) | 0.834 |
| Time 5 | | 0.025 | 49 | 39.39 (5.23) | < 0.001 | 47 | 37.21 (5.09) | 0.721 |
| Overall effect of group | GSES | 0.791 | | | | | | |
| Overall effect of time | | 0.312 | | | | | | |
| Time × group | | 0.331† | | | | | | |
| Baseline | | Ref | 75 | 31.20 (5.44) | – | 71 | 31.41 (4.82) | – |
| Time 1 | | 0.201 | 61 | 31.07 (4.66) | – | 58 | 31.88 (4.99) | – |
| Time 3 | | 0.793 | 57 | 31.56 (5.26) | – | 54 | 31.67 (5.56) | – |
| Time 5 | | 0.123 | 55 | 32.69 (5.46) | – | 52 | 31.83 (4.70) | – |
| Overall effect of group | EQ-VAS | 0.875 | | | | | | |
| Overall effect of time | | 0.209 | | | | | | |
| Time × group | | 0.022‡ | | | | | | |
| Baseline | | Ref | 76 | 68.13 (19.53) | Ref | 71 | 72.44 (15.80) | Ref |
| Time 1 | | 0.038 | 59 | 74.64 (18.05) | 0.026 | 60 | 75.30 (16.36) | 0.501 |
| Time 3 | | 0.256 | 56 | 78.27 (14.84) | 0.007 | 56 | 71.89 (18.33) | 0.195 |
| Time 5 | | 0.462 | 57 | 77.33 (15.52) | 0.031 | 51 | 72.06 (17.19) | 0.137 |
| Overall effect of group | HDL-C | 0.555 | | | | | | |
| Overall effect of time | | 0.002 | | | | | | |
| Time × group | | 0.016‡ | | | | | | |
| Baseline | | Ref | 74 | 1.24 (0.39) | Ref | 75 | 1.30 (0.48) | Ref |
| Time 1 | | 0.075 | 59 | 1.37 (0.43) | 0.003 | 60 | 1.23 (0.36) | 0.781 |
| Time 3 | | < 0.001 | 47 | 1.42 (0.55) | < 0.001 | 44 | 1.27 (0.37) | 0.655 |
| Time 5 | | 0.036 | 58 | 1.39 (0.41) | < 0.001 | 55 | 1.27 (0.42) | 0.459 |
| Overall effect of group | BMI | 0.962 | | | | | | |
| Overall effect of time | | 0.973 | | | | | | |
| Time × group | | 0.061* | | | | | | |
| Baseline | | Ref | 65 | 27.12 (3.91) | Ref | 67 | 27.07 (4.85) | Ref |
| Time 1 | | 0.951 | 58 | 26.83 (3.93) | 0.076 | 56 | 27.83 (4.97) | 0.100 |
| Time 3 | | 0.673 | 43 | 27.33 (4.22) | 0.440 | 44 | 28.04 (4.77) | 0.874 |
| Time 5 | | 0.897 | 53 | 27.11 (4.43) | 0.701 | 52 | 27.78 (5.07) | 0.862 |
| Overall effect of group | Weight | 0.772 | | | | | | |
| Overall effect of time | | 0.885 | | | | | | |
| Time × group | | 0.047‡ | | | | | | |
| Baseline | | Ref | 74 | 81.75 (13.61) | Ref | 74 | 81.44 (17.45) | Ref |
| Time 1 | | 0.739 | 61 | 81.77 (13.58) | 0.085 | 61 | 84.05 (17.19) | 0.045 |
| Time 3 | | 0.607 | 47 | 83.83 (15.52) | 0.788 | 46 | 86.40 (18.64) | 0.372 |
| Time 5 | | 0.426 | 56 | 84.52 (15.16) | 0.864 | 57 | 86.15 (18.20) | 0.408 |
| Overall effect of group | Smoking | 0.029 | | | | | | |
| Overall effect of time | | 0.428 | | | | | | |
| Time × group | | 0.787† | | | | | | |

**Table 2** Continued

| | Outcome | P value | N | PBL group Mean (SD) or n (%) | P value | N | Control group Mean (SD) or n (%) | P value |
|---|---|---|---|---|---|---|---|---|
| Baseline | | Ref | 79 | 7 (8.9) | – | 76 | 8 (10.5) | – |
| Time 1 | | 0.971 | 62 | 4 (6.5) | – | 63 | 9 (14.2) | – |
| Time 3 | | 0.309 | 48 | 5 (10.4) | – | 52 | 9 (17.3) | – |
| Time 5 | | 0.461 | 65 | 3 (4.6) | – | 65 | 7 (10.8) | – |

Time 1=1-year follow-up. Time 3=3-year follow-up. Time 5=5-year follow-up.
Stratified analysis is adjusted age, sex, educational level and marital status. Mixed linear model analyses were performed for (SWE-CES, GSES, EQ-VAS, HDL-C, BMI and Weight) and mixed logistic regression for (smoking).
*Interaction year × group p<0.10. Analysis stratified by group (PBL) and (Control group).
†Interaction year × group p>0.10. Analysis not stratified.
‡Interaction year × group p<0.05. Analysis stratified by groups (PBL) and (control group).
BMI, body mass index; EQ-VAS, EuroQol Visul Analogue Scale ; GSES, General Self-Efficacy Scale; HDL-C, high-density lipoprotein cholesterol; SWE-CES, Swedish Coronary Empowerment Scale.

participants' problem-solving skills, which is not possible with home-sent patient information. Increased patient empowerment can lead to lifestyle changes as well as decisions to make no changes at all. Improvements in patient empowerment in our study indicate a clinically relevant change in for example HDL-C that also consisted as clinically relevant after 5 years follow-up. Our findings confirm that cardiac rehabilitation focusing on behavioural strategies may be beneficial in maintaining healthy levels of HDL-C.

Multidisciplinary weigh loss behavioural interventions in cardiac rehabilitation may be effective in the short run.[41] We did not observe any changes in BMI between or within groups over time. In contrast, a meta-analysis study involving almost 20 000 participants showed that internet-based education significantly reduced cardiac risk factors regarding blood pressure, blood lipids, weight and physical inactivity, with changes lasting up to 1 year.[35] We did not include physical activity which could have been valuable to gain a deeper understanding of the non-significant results of BMI and weight over the 5 years follow-up.

A cognitive nurse-led intervention designed to improve physical activity by using repeated telephone calls and text messaging consultations improved BMI at a 6-month follow-up compared with controls.[42] Our study was also nurse led, but the intervention did not include extra consultations, telephone calls or text message for achieving lifestyle goals. Digital aids with continuous support may be helpful when new health promotion activities are being started.[43 44]

The results from the Euroaspire V study reported a high prevalence of persistent smokers after a CHD event.[13] In this study, PBL did not result in a successful lifestyle change in smoking during the follow-up. Even though few participants were smokers in our study, there was a trend, although not a significant one, of participants in the control group being smokers more often than those in the PBL group. Higher age, low level of education and living alone were also significantly associated with smoking. As reported from other research,

these sociodemographic aspects could be considered in stratifying for cardiac risk factors.[45]

We observed no long-term effects of PBL on self-efficacy. This finding is in contrast to Su *et al*, who reported significant improvements in self-efficacy from an eHealth cardiac rehabilitation intervention.[46] Both studies included cardiac patients and used social cognitive theory methods. However, the inconsistency can probably be explained by the facts that participants in our study were older, several lived alone and the majority had compulsory education. We accounted for these factors, suggesting that future studies may need to take sociodemographic factors into account in studies designed for cardiac rehabilitation interventions.

The COR-PRIM study has some study limitations and strengths that we would like to address. We aimed to include a representative sample of patients with CHD. To enable this, the study sample was based on rigorous inclusion criteria. Participants with CHD that was verified by coronary intervention with PCI, CABG or previous myocardial infarction. Participants were stable in their CHD and were optimally pharmacologically treated in the last month before inclusion. If applicable, patients should have completed cardiac school at the hospital. With these inclusion criteria, we consider that our sample is representative of a population being diagnosed and treated for CHD. However, the majority in this study were retired and therefore the results need to be interpreted with caution according to a younger population with CHD.

Most of the participants in this study were men and the majority were living in suburban areas. This should be considered as limitation to the generalisability of the results of the study. To mitigate the unequal distribution of men and woman, analysis was adjusted for sex.

Another aspect is the high number of participants that did not want to be included. This can be partly due to the reluctance to be in a study and having to schedule meetings and follow-up visits. We have seen this recently in another study in a similar patient group[47] in which the lack of time and other commitments (eg, travel or taking

care of grandchildren) were presented as reasons of non-participation. This may have consequences of the generalisability of the study results.

Moreover, the poor uptake of the PBL programme could be due to the programme design involving 13 sessions for 2 hours over 1 year. This may be regarded as challenging to manage especially if patients reside far from the hospital and not having economic compensation, which was a reason for non-participation in a similar PBL programme including 10 sessions. Of 800 screened patients with rheumatoid arthritis nearly 600 patients did not participate and stated, for example, that they did not want to even if they fit the inclusion criterions.[48] However, patients abandon patient education even if there are few patient education sessions. This was found in the context of diabetes education for 3 days. Only 24% of the patients attended the education.[49] One way to improve barriers to the uptake of the PBL programme is to offer a digital PBL programme. Digital programme enables people to participate despite living in sub-urban areas who do not have practical or economic resources to travel for a PBL programme. The advantage of digital PBL programme is also that selected parts of the programme could be included as prerecorded modules. This could make the programme more flexible and accessible to a broader group of patients with CHD, for example, for those of employable age. We believe that PBL as a pedagogy, closely offered in a digital way to the patients may be a future option.

All participants had similar opportunities to undergo traditional cardiac school for 1 day. Cardiac school may have influenced participants attitude and knowledge and the results in this study. However, the primary outcome in this study was patient empowerment, and completion of cardiac school before the intervention might have low influence on patient empowerment.

It is a challenge to manage problems arising due to losst to follow-up and deaths while performing a study with long-term follow-up design. During a longitudinal study, participants are exposed by several things for example social media, television (TV) campaigns and articles. This affects both control and intervention participants. However, participants in PBL group have learnt to appraise patient information as more or less evident. Publicity in various media can influence the attitude towards important secondary preventive factors such as treatment with statins.[50] It is, therefore, important that the PBL group are supported with an evidence-based approach. Furthermore, participants received a strategy during the 1 year intervention to reflect on public information presented in, for example, newspapers and TV. They also felt empowered in a new way to discuss treatment and self-care in the group.[51]

An increased patient empowerment (SWE-CES) in patients may not result in adherence to guidelines. Most of the patients were retired and cohabiting, which may imply that lifestyles were influenced by partners.[52] They were invited to some of the PBL sessions to take part in discussions about their own questions with healthcare professionals, but we did not follow-up the spouses after the intervention. However, the PBL-intervention was performed for 1 year, and we believe this is a strength, as the education became a part of the patients' lives.

Patients were recruited 6–12 months after the cardiac event on the basis that many continue to smoke, live with hypertension and elevated cholesterol levels about 6 months after starting the medication,[13] which indicate that patients returned to old habits as before the cardiac event. Another clinical observation was that many patients wait for the visit to the cardiologist after discharge from the hospital. However, this is delayed with several months due to heavy workload, leaving the patients to themselves. Thus, we suggest that our intervention fills a gap during the rehabilitation process. A major strength and novelty of this study is that it was performed in primary care after the hospital-based rehabilitation programme. This fact also explains why it was so long time after the event.

A major strength of our study is the long-term follow-up, which is very much needed in interventions aiming at cardiac preventive behavioural changes. Another strength is that the intervention was led by pedagogical trained specialist nurses who worked in primary care.

Patient education led by healthcare professionals skilled in the chosen educational model improve patients' knowledge, empowerment and management of the disease.[53] A further advantage is that patients could choose to participate in group education, but few chose to participate, a digital PBL education may therefore be a future option.

## CONCLUSION

One-year PBL intervention had positive effect on patient empowerment, health status and HDL-C compared with the control group but did not result in improvements in other cardiac risk factors or self-efficacy. Covariates age, sex, education and marital status emerge both as healthy and cardiac risk factors.

**Author affiliations**
[1]Department of Health, Medicine and Caring Sciences, Linköping University, Linköping, Sweden
[2]Centre for Clinical Research Sörmland, Uppsala University, Eskilstuna, Sweden
[3]Primary Health Care Center Vikbolandet, Region Östergötland, Vikbolandet, Sweden
[4]Julius Center, University Medical Center Utrecht, Utrecht, Netherlands

**Acknowledgements** We would like to thank study participants and nurses who contributed to the PBL intervention and for the monitoring during the follow-ups.

**Contributors** CA, PT, SN, TJ and AKK contributed to the conception or design of the work. AKK contributed to the acquisition of the work and CA, PT, SN, TJ, NK and AKK contributed to analysis or interpretation of data for the work. CA drafted the manuscript and AKK, PT, SN, TJ and NK made substantial intellectual contributions to the manuscript. AKK is responsible for the overall content as the guarantor. All authors agreed to be accountable for all parts of the manuscript and approve the publication of the final manuscript.

**Funding** This work was supported by the Swedish Heart and Lung Association project numbers E091/10, E122/11, E083/12 and E103/13 and the Country Council of Region Östergötland, Sweden, project numbers LIO-92281, LIO-125151, LIO-27535, LIO-354951 and LIO-433801.

**Competing interests** None declared.

**Patient and public involvement** Patients and/or the public were not involved in the design, or conduct, or reporting, or dissemination plans of this research.

**Patient consent for publication** Consent obtained directly from patient(s).

**Ethics approval** This study involves human participants and was approved by Regional Ethics Committee of Linköping, Ref. No. Dnr 2010/128-31 Participants gave informed consent to participate in the study before taking part.

**Provenance and peer review** Not commissioned; externally peer reviewed.

**Data availability statement** Data are available on reasonable request.

**Open access** This is an open access article distributed in accordance with the Creative Commons Attribution 4.0 Unported (CC BY 4.0) license, which permits others to copy, redistribute, remix, transform and build upon this work for any purpose, provided the original work is properly cited, a link to the licence is given, and indication of whether changes were made. See: https://creativecommons.org/licenses/by/4.0/.

**ORCID iD**
Christina Andreae http://orcid.org/0000-0002-1482-767X

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
