## [Reviewer comments · BMJ Open]

ARTICLE DETAILS

TITLE (PROVISIONAL)	Does problem-based learning improve patient empowerment and cardiac risk factors in patients with coronary heart disease in a Swedish primary care setting? A long-term prospective, randomized, parallel single randomized trial (COR-PRIM).
AUTHORS	Andreae, Christina; Tingström, Pia; Nilsson, Staffan; Jaarsma, Tiny; Karlsson, Nadine; Köhler, Anita Kärner

VERSION 1 – REVIEW

REVIEWER	Timoteo, Ana Teresa Ctr Hosp Lisboa Cent
REVIEW RETURNED	19-Jun-2022

GENERAL COMMENTS	The authors of the present manuscript sought to evaluate the long-term effects (up to five years) of a one-year problem-based learning (PBL) program on self-management and cardiac risk factors in patients with stable coronary heart disease at a primary care setting in Sweden. This is a prospective, randomized single centre trial and 157 patients were included. Outcomes were assessed with questionnaires of patient empowerment, self-efficacy, self-rated health status and also clinical and laboratory data of cardiovascular risk factors, adjusted for sociodemographic factors. With PBL intervention, they showed that in general, there is a positive long-term effect, particularly in cardiovascular risk factors, patient empowerment and self-rated health. No effect was observed for BMI, weight, smoking and self-efficacy. In practice, worldwide participation in cardiac rehabilitation programs remains low in patients with stable coronary artery disease, and most are hospital-based, which is something that can cause some limitations due to restrictions in the number of patients that can be included in those programs. At primary care level, it is essential to further emphasize to the patient the importance of cardiovascular risk factors control and medication adherence. Empowering the patient for self-management is of particularly importance. Short-term programs of PBL have shown that some improvements can be achieved. However, in medium-term, results are in general disappointing. The present study applied a longer program (up to one year) and outcomes were assessed on long-term. The program consisted of groups sessions (6 -9 participants) in primary care led by nurses who had advanced training in leading problem-based with 13 sessions over one year. The lost to follow up rate is acceptable and the rate of completion of the one-year program was very high. Characteristics of both groups were well balanced. My specific comments are:
---

	1 - Physical activity should have been addressed as a parameter during the intervention and might have explained the reasons for a null effect on BMI and weight 2 - It is of particular relevance that initially, 65% of possible candidates were excluded, mostly because they declined to participate. It is important to identify specific barriers for the implementation of these programs. Most patients included were men, from urban areas and most were retired. This can give us some hints about barriers to the implementation of PBL. 3 - Only patients that had previously completed cardiac rehabilitation were included. This is a factor that can influence results (because patients had some previously knowledge about risk factors control), and it will certainly be a barrier for a more general application of such an intervention, due to the small rate of patients included in cardiac rehabilitation programs. 4 - The manuscript structure requires some improvement. It is very difficult to understand where the transition between sub-topics is. For instance in the outcome definition, this subtitle should be in bold and the definitions underlined? Improvement in the structure is required and mandatory. 5 - Statistical analysis is quite complex and I recommend to be reviewed by an expert in statistics. 6 - Minor English grammar corrections.
--	---

REVIEWER	Svensson, Leif Karolinska Institute, Department of Medicine, Cardiology
REVIEW RETURNED	07-Jul-2022

GENERAL COMMENTS	Long-term effects of problem-based learning on selfmanagement and risk-factors in coronary heart disease – A randomized study in primary care (COR-PRIM) A Swedish study with the objectives to investigate long-term effects of a one-year problem-based learning (PBL) on self-management and cardiac risk factors in patients with coronary heart disease. The design of the study is prospective, randomized in parallel single centre (Linköping) trial. This paper is well written but to me somewhat unusual disponded and maybe more follow quality studies. However, this is for me not a problem. Basically, this type of patient related research is extremely important and I'm therefore very positive to have this paper published. However, before taken this paper into further consideration/decision I have some major and minor issues to discuss with the authors. Please, more deeply discuss the general effect on knowledge/"problem learning" in this field and therefore the impact of your result in regards to newspaper information, TV programe, GP influence and so on.... 1 – 5 years follow up is a long time.
--

My major concern is however the magnitude of eligible and recruited patients in this trial. It is well known that it is difficult to engage patients over time in cardiac rehab. Mostly, only the most motivated patients come to this kind of (in your case extremely ambitious program). So, is your cohort representative to a general cardiac post event population? Or have you recruited only the most motivated patients?

What is reasonable to have as the main outcome in a study like this? Well being?...effect on 5 year survival? Effect on smokers? Or effect on wider riskprofiles such as Low HDL, High LDL, P-Glucose, Bloodpressure? ...please comment on this.

Can you also comment the overall effect on cardiac rehabilitation (is this proven? And if yes on what outcome?) on cardiac rehab program postinfarct / PCI /CABG. This should/must be discussed more profound in the introduction.

What do you mean by "Long term effect" and on what? Is 1 year long term? 3 years? 5 years? You should pic 1 or 3 years as your primary endpoint. In 3-5 years so many other things can and will happening....

We need to know much more in regards to your intervention! How many nurses held those sessions? And did you find any differences in between them in terms of result? Drop outs? Please give a detailed description on every part of the intervention (N=13). I think this should be done as a supplement.

It's a huge intervention: 6-9 patients in each group meeting for 13 times! To me this is not realistic or what do you think? in a daily life situation...? Please comment. Please also comment how you design your intervention? Why 13 meetings and not 20 or 5? And how many minutes every time?

You had 446 eligible patients, 289 were excluded and 246 did not want to be part of this trial. Is this a problem...please discuss this

Furthermore, another difficulty for me is why you waited so long with recruitment? 284 days? After the event? If I understand you right? Motivation....? Please discuss.

Please comment your decision to analyse the piced riskfactors? Why not LDL, BP, FYSS? And other more well known risc factors?

	You should remove a lots of your reported results from the method section to results. Also in the same matter....sample is a result....Power is OK as a section under statistics. Also, from where did you pic your power calculation? I did not understand this? Finally, is this intervention something you can or should recommend other clinics to implement? What will be your next step/next study? In regards to your results?
--	---

REVIEWER	Narayan, Pradeep Bristol Heart Institute
REVIEW RETURNED	26-Aug-2022

GENERAL COMMENTS	This is a well-conducted and well-reported RCT assessing the role of problem-based learning on self-management and risk factors in coronary heart disease. The study has a 5-year follow-up and loss to follow-up was minimal for which the authors must be congratulated. There are a few queries that the authors may wish to clarify. 1. BMI- The authors mention- Page 16- There was no statistically significant main effect (time or group) for BMI (Body Mass Index) Page 17- There was a significant interaction between time and group at the significance level 10% ($p = 0.061$). Therefore, the analysis was stratified by group. Why was a significance level calculated at 10% in this case? Especially since under the statistical analysis the authors mention- The level of significance used was $p < 0.05$. Also, at first glance, the above 2 statements (pages 16 and 17) on BMI appear contradictory and perhaps the authors should try and rephrase or explain it in a little more detail. 2. The inclusion criteria included – Patients diagnosed with CHD verified by percutaneous coronary intervention (PCI) or coronary artery by-pass surgery (CABG) and CABG+PCI or myocardial infarction within 6-12 months prior to the intervention and who had previously completed cardiac rehabilitation. This appears to be a heterogenous group where adherence to either PBL or even standard rehabilitation may be affected based on whether they had a CABG or were just medically managed. The authors have not reported the distribution of the different categories of these patients. Nor have they examined the influence of this baseline variation. It is quite likely that this may have affected the results observed. 3. Most of the patients who were screened but not included in the study were because they declined to participate ($n=246$). This is very unusual for a study where the intervention would be expected to benefit the patients and is “minimal risk”. Do the authors have a feel for the reason behind this? This may be important because PBL even though beneficial seems to have a low acceptance.
---

	4. With regards to HDL-C differences, can the authors confirm that all the patients in the study were treated with Optimal Medical Therapy in both arms? Did they have a protocol for lipid control at the outset?
--	--

REVIEWER	Hong, Kui the Second Affiliated Hospital of Nanchang University, Nanchang of Jiangxi, 330006, China, Department of Cardiovascular Medicine
REVIEW RETURNED	01-Sep-2022

GENERAL COMMENTS	In this manuscript, the authors conducted a randomized controlled study and found that problem-based learning (PBL) intervention in primary care which had positive long-term effects on self management and cardiac risk factors in patients with coronary heart disease (CHD). This topic is meaningful. However, the major concern is that the authors have published one paper (PMID : 32795267) which also highlights the significance of PBL invention on patients with CHD. The same participants and trial registration. The similar design and outcomes. The main difference is the prolonged followed-up years and some positive results in the present study. Thus, please elaborate the purpose and significance of this article in the Introduction and Discussion. Additionally, there are several minor points that they should consider as follows. For GSES (self-efficacy) there was no significant group effect or change over time, but why Table 2 does not show the p-value? Please clarify why the baseline characteristics were so limited? Please explain what is different in your approach as opposed to the earlier approach (PMID : 32795267)? The authors concluded that one-year PBL programme may reduce cardiac risk factors. However, in the whole article, you only observed that HDL-C improved over the five years follow-up. Thus, I suggest the authors should conclude the results with caution and temper. Besides, did the authors assess the other lipid levels such as TC and LDL-C?
---

VERSION 1 – AUTHOR RESPONSE

Reviewer 1	
Dr. Ana Teresa Timoteo, Ctr Hosp Lisboa Cent Comments to the Author:	
The authors of the present manuscript sought to evaluate the long-term effects (up to five years) of a one-year problem-based learning (PBL) program on self-management and cardiac risk factors in patients with stable coronary heart disease at a primary care setting in Sweden. This is a prospective, randomized single centre trial and 157 patients were included. Outcomes were assessed with questionnaires of patient empowerment, self-efficacy, self-rated health	

status and also clinical and laboratory data of cardiovascular risk factors, adjusted for sociodemographic factors. With PBL intervention, they showed that in general, there is a positive long-term effect, particularly in cardiovascular risk factors, patient empowerment and self-rated health. No effect was observed for BMI, weight, smoking and self-efficacy. In practice, worldwide participation in cardiac rehabilitation programs remains low in patients with stable coronary artery disease, and most are hospital-based, which is something that can cause some limitations due to restrictions in the number of patients that can be included in those programs. At primary care level, it is essential to further emphasize to the patient the importance of cardiovascular risk factors control and medication adherence. Empowering the patient for self-management is of particularly importance. Short-term programs of PBL have shown that some improvements can be achieved. However, in medium-term, results are in general disappointing. The present study applied a longer program (up to one year) and outcomes were assessed on long-term. The program consisted of groups sessions (6-9 participants) in primary care led by nurses who had advanced training in leading problem-based with 13 sessions over one year. The lost to follow up rate is acceptable and the rate of completion of the one-year program was very high. Characteristics of both groups were well balanced. My specific comments are: Physical activity should have been addressed as a parameter during the intervention and might have explained the reasons for a null effect on BMI and weight	Thank you for thoughtful comment. We hope we have interpreted your question correctly. We agree, physical activity would have been relevant to measure to explain the results of BMI and weight. We addressed this in the discussion on page 19.
It is of particular relevance that initially, 65% of possible candidates were excluded, mostly because they declined to participate. It is important to identify specific barriers for the implementation of these programs. Most patients included were men, from urban areas and most were retired. This can give us some	We agree and have extended the discussion to meet raised points on page 21.

hints about barriers to the implementation of PBL.	
Only patients that had previously completed cardiac rehabilitation were included. This is a factor that can influence results (because patients had some previously knowledge about risk factors control), and it will certainly be a barrier for a more general application of such an intervention, due to the small rate of patients included in cardiac rehabilitation programs.	Thank you for the important point, we cannot know for sure if previous cardiac school influenced the participants' knowledge and the results in this study. All participants had similar opportunities to undergo traditional cardiac school. However, cardiac school is often limited to one day. Also, the primary outcome was empowerment, and completion of cardiac school may have low influence on this. We have addressed this points in the discussion on page 21 and 22.
The manuscript structure requires some improvement. It is very difficult to understand where the transition between sub-topics is. For instance in the outcome definition, this subtitle should be in bold and the definitions underlined? Improvement in the structure is required and mandatory.	Thank you for the raised point. We have made it clear in the hole manuscript, and findings are now presented separately in blocks.
Statistical analysis is quite complex and I recommend to be reviewed by an expert in statistics.	The statistical analyses were performed by a statistician.
Minor English grammar corrections.	Thank you. The manuscript has been reviewed by native English language reviewer.
Reviewer 2	
Prof. Leif Svensson, Karolinska Institute, Department of Medicine Comments to the Author	
A Swedish study with the objectives to investigate long-term effects of a one-year problembased learning (PBL) on self-management and cardiac risk factors in patients with coronary heart disease. The design of the study is prospective, randomized in parallel single centre (Linköping) trial. This paper is well written but to me somewhat unusual disponed and maybe more follow quality studies. However, this is for me not a problem. Basically, this type of patient related research is extremely important and I´m therefore very positive to have this paper published. However, before taken this paper into further consideration/decision I have	

some major and minor issues to discuss with the authors.	
1. Please, more deeply discuss the general effect on knowledge/" problem learning" in this field and therefore the impact of your result in regards to newspaper information, TV programe, GP influence and so on....1 – 5 years follow up is a long time.	Thank you. We have extended the discussion to meet this point on page 20 and 21.
2. My major concern is however the magnitude of eligible and recruited patients in this trial. It is well known that it is difficult to engage patients over time in cardiac rehab. Mostly, only the most motivated patients come to this kind of (in your case extremely ambitious program). So, is your cohort representative to a general cardiac post event population? Or have you recruited only the most motivated patients?	Thank you for an important question. We have no evidence that the most motivated patients gave consent to the study. All persons assessed for eligibility criteria were invited to the study with careful inclusion and exclusion criteria which counteracted some of these possible barriers. We have created the most possible conditions to prevent bias of eventually motivated patients. We can with some caution suggest that the results are representative to a general cardiac population.
3. What is reasonable to have as the main outcome in a study like this? Well being?...effect on 5 year survival? Effect on smokers? Or effect on wider riskprofeiles such as Low HDL, High LDL, P-Glucose, Bloodpressure? ...please comment on this.	An intervention that compares two different educational interventions (PBL versus home sent information) has pedagogy in focus and then knowledge and understanding, skills and abilities as well as values and attitudes (learning outcomes) are in the foreground, i.e., empowerment is relevant to choose as the main outcome. Empowerment means for example the ability of people to achieve goals, overcome obstacles, use strategies for self-care choices, which we intended to achieve. We sought to influence patient empowerment and then the secondary is the lifestyle change itself. Survival is hard to get sufficient power for. We would probably have needed considerably more participants in each group.
4. Can you also comment the overall effect on cardiac rehabilitation (is this proven? And if yes on what outcome?) on cardiac rehab program postinfarct / PCI /CABG. This should/must be discussed more profound in the introduction.	We agree this is lacking and have meet the raised points in the introduction on page 3 and 4.
5. What do you mean by "Long term effect" and on what? Is 1 year long term? 3 years? 5 years? You should pic 1 or 3 years as your primary endpoint. In 3-5 years so many other things can and will happening...	Results from one-year follow up were published previously (reference 19). We considered it illustrative to show what happens after 3 as well as 5 years in the same analyses e.g., shown in table 2 and boxplots (Figure 2.). Maybe this is a little unusual but should be considered rather as a strength of the study.

	In addition, longitudinal designed patient education studies with carefully described methods and intervention are being requested.
6. We need to know much more in regards to your intervention! How many nurses held those sessions? And did you find any differences in between them in terms of result? Drop outs? Please give a detailed description on every part of the intervention (N=13). I think this should be done as a supplement.	Altogether, there were six nurses where of five were district nurses and one had long experience of working in an out-patient clinic at a hospital caring for patients with CHD during the cardiac rehabilitation. We did not evaluate the nurses work in terms of patients results or regarding dropouts. Please, also see response to Associate editor on page 2 and 3. The intervention has been clarified and will be attached as a supplementary. There was no statistically significant difference in the dropout rate between treatment groups as tested by chi-squared test ($p=0.980$). We added the results of chi-squared test on page 11.
7. It's a huge intervention: 6-9 patients in each group meeting for 13 times! To me this is not realistic or what do you think? in a daily life situation...? Please comment. Please also comment how you design your intervention? Why 13 meetings and not 20 or 5? And how many minutes every time?	The intervention worked well because the majority were seniors with easier access to attend the intervention. There were 13 meetings (2 hours per meeting). This was based on the multiple of areas to discuss for life-styles changes in coronary disease. Furthermore, lifestyles changes require work over time. To be feasible in the patient's life, a long training is required. The PBL model was validated in a previous study reference 21. Tingström P, Kamwendo K, Göransson A, et al. Validation and feasibility of problem-based learning in rehabilitation of patients with coronary artery disease. Patient Educ Couns 2002;47:337-345.doi:10.1016/S0738-3991(02)00007-1 Information is added in the method on page 7.
8. You had 446 eligible patients, 289 were excluded and 246 did not want to be part of this trial. Is this a problem...please discuss this	We agree that we had a high number of participants that did not want to be included. This can be partly due to the reluctance to be in a study and having to schedule meetings and follow up visits. We have seen this recently in another study in a similar patient group in which the lack of time and other commitments (e.g., travel or taking care of

	grandchildren) as main reasons of non-participation. We realize this has consequences of the generalizability of the study results. We discussed the raised points on page 22.
9. Furthermore, another difficulty for me is why you waited so long with recruitment? 284 days? After the event? If I understand you right? Motivation....? Please discuss.	The inclusion criteria were decided within the research group and in alliance with clinicians at one hospital. Motives for recruiting the patients 6-12 months after the cardiac event was that many patients continue to smoke, live with hypertension and elevated cholesterol levels about six months after starting the medication which indicate that they returned to old habits as before the cardiac event. Another clinical observation was that many patients wait for the visit to the cardiologist after discharge from the hospital. However, this is delayed with several months due to heavy workload, leaving the patients to themselves. Thus, we believe that our intervention fills a gap during the rehabilitation process. A major strength and novelty of this study is that it was performed in primary care after the hospital-based rehabilitation program. This fact also explains why it was so long time after the event. We have extended the discussion on page 21.
10. Please comment your decision to analyse the piced riskfactors? Why not LDL, BP, FYSS? And other more well known risc factors?	Total cholesterol, LDL, triglycerides, systolic and diastolic BP are presented in reference 19, table 3 on page 8. In this study, cardiac risk factors that came out significant after one year follow up were selected for the analyses after 3 and 5 years i.e., body weight, body mass index and HDL-C. We considered that behavioural changes may take time and therefore we selected smoking and behavioural indicators self-efficacy that was non-significant at 1 year follow-up.
11. You should remove a lots of your reported results from the method section to results.	We agree and have removed the results of the sample description to the results on page 10-11.
12. Also in the same matter....sample is a result....Power is OK as a section under statistics. Also, from where did you pic your power calculation? I did not understand this?	We used the SWE-DES 23 scale (empowerment instrument) to determine the sample size in the groups. The SWE DES-scale was earlier used in patients with diabetes which we considered comparable with CHD in terms of life-long disease.

	In our study, the mean value for the CHD-patients in the control group were expected to be 30 (SD=12) whilst the analogue value for those randomized to PBL was 36 (same SD=). At a significant level of 5% and a power of 80% a required sample size was 63/group. However we enrolled 157 patients e.g., 78 resp 79 patients /group to manage losses to follow-up.
13. Finally, is this intervention something you can or should recommend other clinics to implement?	By strengthening empowerment, it may be possible to change lifestyle habits - PBL in a group provides support in a socio-cultural perspective to learn together. Therefore, a tutor is needed to support the learning process. In a primary care perspective, it could be possible to implement PBL in cardiac rehabilitation, but to consider other populations for PBL new tests need to be performed.
14. What will be your next step/next study? In regards to your results?	With regard to that several declined to participate. One possible explanation could be the challenges of combining work and education. We contacted the social security office to give participants the opportunity to participate in PBL during working hours, but this was not possible. There are other practical obstacles to arranging physical meetings, such as pandemics or geographical barriers if you live in sub-urban areas. We now see an opportunity to develop a digital PBL program and plan to conduct a small-scale feasibility study for people with mental barriers conducting physical activity after an event of CHD starting in 2023.
Reviewer 3	
Dr. Pradeep Narayan, Bristol Heart Institute Comments to the Author	
This is a well-conducted and well-reported RCT assessing the role of problem-based learning on self-management and risk factors in coronary heart disease. The study has a 5-year follow-up and loss to follow-up was minimal for which the authors must be congratulated. There are a few queries that the authors may wish to clarify.	
1. BMI- The authors mention-	Thank you for the comment. The statistical analysis of main effects (time and group) for BMI was performed at the 5% level of

Page 16- There was no statistically significant main effect (time or group) for BMI (Body Mass Index) Page 17- There was a significant interaction between time and group at the significance level 10% ($p = 0.061$). Therefore, the analysis was stratified by group. Why was a significance level calculated at 10% in this case? Especially since under the statistical analysis the authors mention- The level of significance used was $p < 0.05$. Also, at first glance, the above 2 statements (pages 16 and 17) on BMI appear contradictory and perhaps the authors should try and rephrase or explain it in a little more detail.	significance as described in “Statistical analysis”. A sensitivity analysis was performed at the significance level of 10% to investigate the significance of the interaction between time and group in an explorative way. Both analyses are now presented in two different pages. We have clarified the results and findings are now presented separately in blocks, please see page 16-18.
2. Most of the patients who were screened but not included in the study were because they declined to participate ($n=246$). This is very unusual for a study where the intervention would be expected to benefit the patients and is “minimal risk”. Do the authors have a feel for the reason behind this? This may be important because PBL even though beneficial seems to have a low acceptance.	Thank you, similar question was asked from reviewer 2. Please see authors response to reviewer 2 question 8.
3. With regards to HDL-C differences, can the authors confirm that all the patients in the study were treated with Optimal Medical Therapy in both arms? Did they have a protocol for lipid control at the outset?	We identified the patient’s lipid levels at baseline using at protocol which was documented by the nurse at the outpatient clinic. About 1/3 of the patients were at baseline treated with lipid lowering medication and should be equally distributed in both arms regarding the randomization procedure. The inclusion criteria involved that their cardiac medication was optimized and not substantially changed during the last month.
Reviewer 4	
Dr. Kui Hong, the Second Affiliated Hospital of Nanchang University, Nanchang of Jiangxi, 330006 China	
In this manuscript, the authors conducted a randomized controlled study and found that problem-based learning (PBL) intervention in primary care which had positive long-term effects on self-management and cardiac risk factors in patients with coronary heart disease (CHD). This topic is meaningful. However, the major concern is that the authors have published one paper (PMID : 32795267)	We agree and have extended the introduction on page 5. This study is elaborated in accordance with the COR-PRIM study basic aim, which was to discover whether PBL provided in primary health care for one year has long-term effects on patient empowerment and self-care, assessed at baseline and, at one, three, and five years after randomization). Thus, in this five

which also highlights the significance of PBL invention on patients with CHD. The same participants and trial registration. The similar design and outcomes. The main difference is the prolonged followed-up years and some positive results in the present study. Thus, please elaborate the purpose and significance of this article in the Introduction and Discussion. Additionally, there are several minor points that they should consider as follows.	year- and final assessment we wanted to identify if the findings in the 1-year follow-up remained or changed, reference 19. By this performance, this article is examining the sustainability of the effects by PBL, which to our knowledge has not been performed before. Long-term data is asked for in other research on effectiveness of patient education about how to lessen risk factors after CHD, this is addressed on page 18 in the discussion.
1. For GSES (self-efficacy) there was no significant group effect or change over time, but why Table 2 does not show the p-value?	We agree that for GSES (self-efficacy) there was no significant group effect (P=0.791) or change over time. We have added the p-value for "Overall effect of time" for GSES (P=0.312) in table 2. For purpose of clarity, we have added the p-value for "Overall effect of time" for all variables in Table 2.
2. Please clarify why the baseline characteristics were so limited?	To counteract the duplication of baseline characteristics we decided to refer some part of baseline characteristics to our previous study which we referred to reference 19.
3. Please explain what is different in your approach as opposed to the earlier approach (PMID : 32795267)?	The approach in this study was to evaluate the long-term effects (3-5 years after the intervention) on patient empowerment (PE) and to follow-up risk factors (weight, BMI and HDL cholesterol), which seemed to be positively impacted on in the 1-year follow up study. We hypothesized that PBL empowered the patients to change self-care compared with standardized home-sent information and we conclude that PBL resulted in long-term effects on patient empowerment, health status and HDL. A result which we believe is relevant as the randomization procedure was thoroughly performed and the patient baseline characteristics did not differ between the groups.
4. The authors concluded that one-year PBL programme may reduce cardiac risk factors. However, in the whole article, you only observed that HDL-C improved over the five years follow-up. Thus, I suggest the authors should conclude the results with caution and temper. Besides, did the authors assess the other lipid levels such as TC and LDL-C?	Thank you, we agree to this point and have adjusted on page 22. We selected study variables that were significant after 1 year of follow-up which explain the lack of other risk factors in the study.

VERSION 2 – REVIEW

REVIEWER	Timoteo, Ana Teresa Ctr Hosp Lisboa Cent
REVIEW RETURNED	17-Nov-2022

GENERAL COMMENTS	My concerns were properly addressed. I do not know if the journal requested an independent statistical review of the manuscript, as it was my recommendation. I strongly suggest that before final approval.
--

REVIEWER	Narayan, Pradeep Bristol Heart Institute
REVIEW RETURNED	22-Nov-2022

GENERAL COMMENTS	Thank you for revising the manuscript. The study confirms that PBL positively contributes to the improvement in patient empowerment, health status, and HDL-C compared to the control group. However, one major issue the study highlights are the poor uptake of PBL. The authors have highlighted this issue but apart from acknowledging the authors could try and give some insight into ways to improve the uptake of PBL. However, this is not mandatory and is only a suggestion.
--

REVIEWER	Hong, Kui the Second Affiliated Hospital of Nanchang University, Nanchang of Jiangxi, 330006, China, Department of Cardiovascular Medicine
REVIEW RETURNED	24-Nov-2022

GENERAL COMMENTS	In the revised manuscript, the authors tried to explain the issues raised by the reviewers, but I still have two questions. 1. About the cardiac risk factors of the study, the authors explained the significant results in the first year (HDL-C, BMI and smoking) were used as secondary outcome, however it may not be enough. If the authors think that empowerment is useful for patients with coronary heart disease to choose self-care strategies, overcome obstacles and change cardiac risk factors, please discuss whether other cardiac risk factors (for instance, blood pressure, LDL-C and blood glucose) were improved after a 5 years follow-up, because the SWE-CES was not significant after 1 year but significant after 5 years. Whether these risk factors were measured after 3 years and 5 years? 2. On page 21 and 22, the authors explained the high rejection rate of research invitations, a study found that lack of free time, travel or taking care of grandchildren were the possible reasons. According to the study they cited, they did not discuss the effects of age and symptom severity, which affect the representation of the study population.
---

VERSION 2 – AUTHOR RESPONSE

Reviewer: 1	
Dr. Ana Teresa Timoteo, Ctr Hosp Lisboa Cent Comments to the Author: My concerns were properly addressed.	
I do not know if the journal requested an independent statistical review of the manuscript, as it was my recommendation. I strongly suggest that before final approval.	We understand if the journal choose to follow the reviewer's recommendation. We are confident in the analyses that have been performed by a professional statistician.
Reviewer: 3	
Dr. Pradeep Narayan, Bristol Heart Institute Comments to the Author:	
Thank you for revising the manuscript. The study confirms that PBL positively contributes to the improvement in patient empowerment, health status, and HDL-C compared to the control group. However, one major issue the study highlights are the poor uptake of PBL. The authors have highlighted this issue but apart from acknowledging the authors could try and give some insight into ways to improve the uptake of PBL. However, this is not mandatory and is only a suggestion.	Thank you, we have expanded our thoughts about digital PBL programme. One way to improve barriers to the uptake of the PBL programme is to offer a digital PBL programme. Digital programme enables people to participate despite living in sub-urban areas who do not have practical or economic resources to travel for a PBL programme. The advantage of digital PBL programme is also that selected parts of the programme could be included as pre-recorded modules. This could make the programme more flexible and accessible to a broader group of patients with CHD, for example for those of employable age. We believe that PBL as a pedagogy, closely offered in a digital way to the patients may be a future option. Please see page 21
Reviewer: 4	
Dr. Kui Hong, the Second Affiliated Hospital of Nanchang University, Nanchang of Jiangxi, 330006 China. Comments to the Author:	

In the revised manuscript, the authors tried to explain the issues raised by the reviewers, but I still have two questions. 1. About the cardiac risk factors of the study, the authors explained the significant results in the first year (HDL-C, BMI and smoking) were used as secondary outcome, however it may not be enough. If the authors think that empowerment is useful for patients with coronary heart disease to choose self-care strategies, overcome obstacles and change cardiac risk factors, please discuss whether other cardiac risk factors (for instance, blood pressure, LDL-C and blood glucose) were improved after a 5 years follow-up, because the SWE-CES was not significant after 1 year but significant after 5 years. Whether these risk factors were measured after 3 years and 5 years?	Thank you for an important and thoughtful comment. We can confirm that SWE-CES was not significantly changed at the 1-year follow-up compared to baseline (ref 19). Achieving behavioral change requires time for reflection and time to create and consolidate behavioral change. The research team decided by consensus to primarily study the effects of the PBL intervention based on previous evidence where weight, BMI and HDL-C were changed at 1 year follow-up compared to baseline, intervention and control groups (ref 19). In addition to these objective outcome variables and the study's primary outcome measures, the research team also included subjective outcome measures to evaluate the effects of the PBL intervention on empowerment and risk factors for CHD. The research team decided to study the impact of the PBL intervention on smoking, which has strong scientific support for the development of CHD, and we also evaluated health status. These factors are important puzzle pieces for understanding how PBL can contribute to empowerment and lifestyle changes to reduce cardiac risk factors in people with CHD. The reviewer asked if other cardiac risk factors (for instance, blood pressure, LDL-C and blood glucose) were improved after a 5 year follow-up, because the SWE-CES was not significant after 1 year but significant after 5 years. Also, whether these risk factors were measured after 3 years and 5 years? These factors were measured at baseline and after 1, 3 and 5 years (17). Although, these factors were not subject to analyze in the present study based on the above explanation.
2. On page 21 and 22, the authors explained the high rejection rate of research invitations, a study found that lack of free time, travel or taking care of grandchildren were the possible	Thank you, this question connects to the one posed by the editor and has been discussed on page 20 and 21.

reasons. According to the study they cited, they did not discuss the effects of age and symptom severity, which affect the representation of the study population.	
--	--

VERSION 3 – REVIEW

REVIEWER	: Hong, Kui the Second Affiliated Hospital of Nanchang University, Nanchang of Jiangxi, 330006, China, Department of Cardiovascular Medicine
REVIEW RETURNED	09-Jan-2023
GENERAL COMMENTS	Thanks to the authors for response and revising the manuscript, I have no further questions.